# Edaravone-Loaded Alginate-Based Nanocomposite Hydrogel Accelerated Chronic Wound Healing in Diabetic Mice

**DOI:** 10.3390/md17050285

**Published:** 2019-05-11

**Authors:** Ying Fan, Wen Wu, Yu Lei, Caroline Gaucher, Shuchen Pei, Jinqiang Zhang, Xuefeng Xia

**Affiliations:** 1Chongqing Key Laboratory of Natural Product Synthesis and Drug Research, School of Pharmaceutical Sciences, Chongqing University, Chongqing 401331, China; fanying@cqu.edu.cn (Y.F.); j.zhang1983@cqu.edu.cn (J.Z.); 2School of Chemistry and Chemical Engineering, Chongqing University of Science and Technology, Chongqiang 401331, China; backer7_leiyu@163.com (Y.L.); peishuchen928@163.com (S.P.); 3Faculté de pharmacé, Université de Lorraine, CITHEFOR F-54000 Nancy CEDEX, France; caroline.gaucher@univ-lorraine.fr

**Keywords:** edaravone, nanocomposite alginate hydrogel, chronic wounds, oxidative stress, reactive oxygen species

## Abstract

Refractory wound healing is one of the most common complications of diabetes. Excessive production of reactive oxygen species (ROS) can cause chronic inflammation and thus impair cutaneous wound healing. Scavenging these ROS in wound dressing may offer effective treatment for chronic wounds. Here, a nanocomposite hydrogel based on alginate and positively charged Eudragit nanoparticles containing edaravone, an efficient free radical scavenger, was developed for maximal ROS sequestration. Eudragit nanoparticles enhanced edaravone solubility and stability breaking the limitations in application. Furthermore, loading these Eudragit nanoparticles into an alginate hydrogel increased the protection and sustained the release of edaravone. The nanocomposite hydrogel is shown to promote wound healing in a dose-dependent way. A low dose of edaravone-loaded nanocomposite hydrogel accelerated wound healing in diabetic mice. On the contrary, a high dose of edaravone might hamper the healing. Those results indicated the dual role of ROS in chronic wounds. In addition, the discovery of this work pointed out that dose could be the key factor limiting the translational application of antioxidants in wound healing.

## 1. Introduction

Diabetes mellitus is one of the most widespread chronic diseases all over the world. Impaired wound healing is a complication of diabetes which results in a failure of chronic wounds (for instance, diabetic foot ulcers) to heal completely. Wound healing is a complex physiological response proceeding with four interrelated dynamic and overlapping phases: coagulation, inflammatory, proliferative, and remodeling phases. The inflammation phase involves the establishment of homeostasis and inflammation; proliferative phase follows with granulation, contraction, and re-epithelialization [1]. The final remodeling phase involves the formation of cellular connective tissue and strengthening of newly formed epithelium [1]. Elevated and sustained reactive oxygen species (ROS) have been detected in vivo and have been associated with impaired wound repair in chronic non-healing wounds [2]. Excessive ROS can directly or indirectly degrade extracellular matrix proteins via proteolysis activation [3]. However low-level ROS play a pivotal role in the normal wound healing response. They act as secondary messengers to recruit many immunocytes and non-lymphoid cells to the wound site and promote tissue repair [2]. Thus the manipulation of ROS presents a promising avenue for accelerating wound healing when they are stalled.

Different drugs and delivery systems have been extensively investigated targeting the phases of wound healing mentioned above. The presently available wound medication requires frequent dressing changes where the wounds have an increased risk of infection until the skin heals completely. Particularly, in the case of chronic wounds, where there is insufficient blood flow and local edema, healing of wounds is very difficult without any active treatments. Also, the current treatment options are expensive, limited, and inefficient which led to the development of new therapeutics to satisfy the unmet clinical needs. 

Under the complex conditions of wound healing, the continuing inflammatory cascade may result in increasing tissue destruction and necrosis rather than healing [4]. Therefore, time-lasting inflammation is a common complication in chronic wound healing process. Topical applications of antioxidants will be useful against oxidative damage and benefit to accelerate the healing of wounds. Edaravone (3-methyl-1-phenyl-2-pyrazolin-5-one) is approved for the treatment of acute cerebral infarction. The protective effect of edaravone against brain damage after ischemia-reperfusion injury is mediated by its ability to scavenge hydroxyl radicals (^•^OH) [5,6]. Edaravone was shown to inhibit postischemic increases in ^•^OH production and tissue injury in the penumbral or recirculated area in rat cerebral ischemia models [5]. In addition, the clinical use of edaravone is well established and has led to satisfying outcomes in cerebral infarction [7,8]. Although edaravone possesses potent free radical scavenging ability, which may promote wound healing, the poor stability and solubility limited its topical applications. 

Alginate is a biopolymer that is naturally occurring, anionic, and it is obtained from brown seaweed [9]. It is linear polysaccharide composed of 1,4-linked β-d-mannuronate and 1,4-linked α-l-guluronate in varying proportions, which is readily biocompatible, non-immunogenic and biodegradable. Alginate dressing absorbs wound fluid maintaining a moist physiological environment and minimizing bacterial infections at the wound site. Maintenance of a moist or wet wound environment has been shown to promote re-epithelialization and reduce scar formation. In addition, alginate hydrogel could serve as the matrix for the aggregation of platelets and erythrocytes to promote wound healing [10]. In view of the advantages of edaravone and alginate hydrogel, a novel edaravone loaded alginate hydrogel is desirable for chronic wound healing. However, it is challenging to achieve the homogenous distribution of water-insoluble edaravone in a hydrogel and sustain release at the target site. In this request, the present work is the first one to present novel alginate based gel-core composites loaded with edaravone polyacrylic ester nanoparticles. This system was designed in order to combine the positive effects of polyacrylic ester nanoparticles in increasing solubility and loading efficiency with the long-lasting release of therapeutics [11]. In addition, alginate was selected as a gel matrix with unique viscoelastic and wound healing properties. The physicochemical characteristics of developed nanocomposite hydrogel were evaluated. In addition, the potential of novel nanocomposites to promote edaravone delivering to wound sites and enhance wound healing were also assessed in diabetic mice (Scheme 1). Considering the dual role of ROS in wound healing, we hypothesize that the dose of edaravone may affect the outcomes. To demonstrate our hypothesis, the dose-response relationship of edaravone related formulations were further evaluated.

## 2. Results

### 2.1. Preparation and Characterization of EDA-NP/EDA-NP-gel

Edaravone-loaded nanoparticles (EDA-NP) were successfully prepared by solvent displacement and evaporation method. The average size represented by the hydrodynamic diameter of the NP (Table 1) was 80.56 ± 14.50 nm for blank-NP and 105.55 ± 1.73 nm for EDA-NP. The polydispersity index was 0.257 ± 0.01 for blank-NP and 0.240 ± 0.02 for EDA-NP. The mean zeta potential was +7.05 ± 4.49 mV for blank-NP and +6.31 ± 3.16 mV for EDA-NP. Edaravone was entrapped within the NP with 51.77% of encapsulation efficiency. Scanning electron microscopy results showed nanoparticles with spherical shape and good dispersibility with sizes less than 100 nm. The nanoparticles are evenly distributed in alginate hydrogel. (Figure 1)

### 2.2. In Vitro Release Kinetics

To mimic the slightly acidic pH environment at wounds [12], the release kinetics of free EDA, EDA-NP, and EDA-NP-gel were performed in phosphate buffer (pH 5) at 37 ℃. An initial burst release phase and a plateau were observed (Figure 2). In the first 0.5 h, about 87% edaravone was released in the free edaravone (Free-EDA) group. In contrast, edaravone-loaded nanoparticles and edaravone-loaded nanocomposite hydrogel presented slow and sustained release profiles, with 53% ± 2% edaravone released from EDA-NP and 48% ± 1% edaravone released from EDA-NP-gel within 0.5 h. In the first 2 h, almost all edaravone was completely released from the Free-EDA group. Subsequently, edaravone slowly degraded until 9 h. In the first 1.5 h, about 84% edaravone was released from EDA-NP, and kept stable until 9 h. EDA-NP-gel displayed a slower release profile, in the first 3 h, about 83% edaravone was released from EDA-NP-gel, and kept stable until 9 h. 

### 2.3. Solubility of Edaravone-Loaded Nanoparticles

The water solubility of edaravone was found to be 1.739 ± 0.04 mg/mL (Figure 3). There was no considerable difference observed in aqueous buffers with different pH. Thus, compared with the solubility of edaravone in water, the encapsulation of edaravone in the nanoparticles increased the solubility to 5 mg/mL, which significantly improved the solubility of EDA.

### 2.4. In Vivo Wound Healing

Wound healing effects of edaravone loaded formulations were evaluated by calculating the wound closure (%) through the measurement of the wound area. The effects of edaravone related topical formulations on wound healing were evaluated in a splinted mouse full thickness excisional model. The wounds were immediately treated with vehicle, EDA-PBS, EDA-NP, and EDA-NP-gel. Digital images of representative wounds of each group from day0, 5, 10, and 13 were presented in Figure 4A. Wound area was tracked over a period of 13 days (Figure 4B). On the 5th post-wounding day (Figure 4C), low dose of edaravone loaded nanocomposite hydrogel (EDA-NP-gel (L)) showed a significantly accelerated repair of wounds compared to that of both diabetic control and normal control group. We found that the wound area of EDA-NP-gel (L) is almost one-third of untreated diabetic wounds and half of free edaravone treated wounds. On the 10th post-wounding day, 96.6% of wounds were repaired with the treatment of EDA-NP-gel (L), which was similar as a wound in untreated normal mice. On the contrary, neither a high dose of edaravone loaded nanocomposite hydrogel (EDA-NP-gel (H)) nor nanoparticles (EDA-NP) could promote wound healing (Figure 5A,B). On the 5th post-wounding day (Figure 5C), a high dose of edaravone loaded formulations, both nanoparticles and nanocomposite hydrogel, worsen the wounds compared to the diabetic control group. The wound healing rates of EDA-NP-gel (H) and EDA-NP are about 30% less than that of untreated diabetic wounds. As presented in Figure 6, EDA-NP-gel (L) significantly improved the wound healing, compared to EDA-NP-gel (H). The wound healing rate of EDA-NP-gel (L) is 3-fold higher than that of EDA-NP-gel (H) on the 5th post-wounding day (Figure 6C). In addition, only a low dose of edaravone loaded nanocomposite hydrogel restored the wound repair ability of diabetic mice close to that of normal mice. 

## 3. Discussion

Time-lasting oxidative stress is the main mechanism for the difficulty in healing wounds of diabetes. Oxidative stress is produced by a large amount of ROS that mediates inflammatory damages and further impedes wound healing. Therefore, ROS scavengers could promote wound healing by mitigating oxidative stress. Numerous novel investigations are being conducted to develop more efficient systems of drug delivery for wound healing activity. In the current study, we introduced edaravone loaded nanocomposite hydrogel as a potential formulation for an efficient wound healing in diabetics. The rationale behind the use of alginate based nanocomposite hydrogel as a drug carrier is to deliver edaravone topically in a sustainable way from its matrix structure and at the same time using the therapeutic potential of alginate in wound healing.

Edaravone has been demonstrated to be a promising substance to scavenge •OH radicals and to inhibit both •OH-dependent and •OH-independent lipid peroxidation. Additionally, it has inhibitory effects on both water- and lipid-soluble peroxyl radical-induced peroxidation systems [13]. We have prepared edaravone loaded Eudragit nanoparticles by solvent displacement and evaporation methods that results in particles with high edaravone encapsulation (~51.77%). Compared with liposomes loading edaravone with an encapsulation efficiency of 23% [14], our developed Eudragit nanoparticles significantly increased the entrapment efficiency. It was reported that the phenol-like structure of edaravone and weak acidic nature having pKa value of 7 allows its ionization in the basic environment [6]. Therefore, we detected the solubility in neutral and acidic environments. EDA-NP has enhanced the water solubility of edaravone by 2.8 times and offered protection against photosensitivity. Loading these EDA-NP into alginate matrix improved the protection by avoiding direct contact with unfriendly factors. Thus some of the limitations of edaravone application have been overcome. Similarly, to improve its physicochemical characteristics including solubility, stability, dissolution, and permeability, the strategy of complexation with hydroxypropyl-β-cyclodextrin was studied by Zeng et al. [15]. By improving solubility, dissolution, and permeability, the hydroxypropyl-β-cyclodextrin-based formulation presented a 10.3-fold improvement in bioavailability [15]. 

Scanning electron microscopy results showed that the nanoparticles with spherical shape and good dispersibility, and the particle size was less than 100 nm. Compared with the blank calcium alginate hydrogel, the surface of EDA-NP-gel was much rougher and distributed with a lot of nanoparticles (Figure 1). The embedded positively-charged nanoparticles interact with negatively-charged sodium alginate through electrostatic interactions, which increased cross-linking and thus reduced the calcium concentration needed for gelation. In agreement with Navarro-Requena et al., low concentration of calcium is a benefit to wound healing, however, high concentration might reduce the activity [16]. In our system, the low concentration of calcium chloride in the nanocomposite hydrogel resulted in a more compact hydrogel with better formability and was good to wound healing. When the calcium ions in the hydrogel are replaced by sodium ions of the wound exudate, the structure of the hydrogel becomes loose and more fluid, which provide a fast on-demand release of edaravone. The addition of glycerin ensures that the nanocomposite hydrogel maintains a moist environment at the wound bed [17]. 

In vitro drug release of edaravone related formulations were performed as it is important for an efficient drug delivery system. EDA-NP and EDA-NP-gel displayed sustained release of edaravone over a period of 9 h (Figure 2). Edaravone released from nanocomposite hydrogel presented a longer sustained release pattern, due to the complex structure of the matrix. Within the nanocomposite hydrogel system, edaravone should cross two “physical barriers” [18]: the inner Eudragit nanoparticles and the outer alginate hydrogel. Moreover, Free-EDA started to degrade from 3 h, while EDA-NP and EDA-NP-gel prevented the degradation within 9 h. 

The activities of edaravone related formulations on wound healing were evaluated in a splinted mouse full thickness excisional model [19]. By performing the in vivo experiments with all the specimens we attempted to provide comparative validation of their potential to accelerate wound closure. We found while measuring the wound area that the EDA-NP-gel (L) showed a two-fold higher wound healing activity compared to that of free edaravone or untreated diabetic wounds and by the 10th post-wounding day, EDA-NP-gel (L) showed a nearly complete healing. In agreements with Koizumin et al. and Naito et al., edaravone solution is effective against burn injury [20] or diabetic wounds [21] with extensive ROS production. Our nanocomposite hydrogel protected and sustained release constant dose of edaravone to fight against extensive ROS production at the diabetic wounds. Interestingly from day 6 of post-wounding, EDA-NP-gel (L) showed an equal potential of wound healing with wounds in untreated normal mice suggesting that the release of edaravone from EDA-NP-gel (L) might down regulate the ROS level close to that in normal mice. The results proved that the antioxidant capacity of edaravone could be used topically to accelerate wound healing in diabetics. To study the dose-response effect of edaravone in diabetic wounds, we increased the loading of edaravone in nanocomposite hydrogel to treat wounds in diabetic mice. However, increasing edaravone was not conducive to wound healing, probably because a high dose of edaravone eliminated too much ROS to hamper the healing process. Compared to EDA-NP-gel (L), EDA-NP-gel (H) released more edaravone and kept the constant release speed within 9 h (Appendix A). Under the physical condition, in the normal wounded site, inflammatory cells produce ROS via the oxidative burst, catalyzed by NADPH oxidase to eliminate the pathogens and promote the inflammatory response [22]. Thus eliminating too much ROS may delay the wound repair. To test our hypothesis, we applied a low dose of edaravone to treat wounds in normal mice where there is no sustained ROS overproduction. We found that edaravone impeded the healing process. On the 5th post-wound day, the wound area is almost twice the size of the untreated wounds in normal mice (Appendix A). On 13th post-wound day, only 72% of wounds were repaired, but untreated normal wounds were repaired completely (Appendix A). These discoveries indicated topical application of edaravone in normal mice might extend the repair cycle. Although free edaravone and EDA-NP show wound healing due to their respective therapeutic activities, EDA-NP-gel accelerated the healing more profoundly than free edaravone and EDA-NP due to the additive activity of encapsulated edaravone and the add-on effect of alginate. 

## 4. Materials and Methods 

### 4.1. Materials

Edaravone (MCI-186; [3-Methyl-1-1-phenyl-2-pyrazolin-5-one]) and trifluoroacetate was purchased from Adamas Co., Ltd (Shanghai, China). Eudragit RL PO was kindly donated by Evonik industries (Shanghai, China). Pluronic F-68, Streptozotocin, sodium dihydrogen phosphate, and alginic acid sodium were purchased from Sigma–Aldrich (St. Louis, USA) Acetone, glycerin, and calcium chloride were purchased from Kelong Chemical Co., Ltd (Chengdu, China). The methanol and acetonitrile were of HPLC grade, whereas the other reagents were of analytical grade.

Eight to nine-weeks-old male C57BL/6J mice weighing 22 g to 25 g were obtained from Sibefu Biotechnology Co., Ltd. (Beijing, China). All animals were maintained in air-conditioned rooms (temperature: 25 ± 0.5 °C; humidity: 50% ± 5%) with a 12 h light–dark cycle. Animals had free access to food and drinking water. All animal experiments were approved by the Third Military Medical Association for Animal Ethics. 

### 4.2. Preparation of Nanoparticles 

Edaravone-loaded NP (EDA-NP) were prepared by solvent displacement and solvent evaporation method. Briefly, 500 mg Eudragit RL 100 was dissolved in 3 mL acetone and 100 mg edaravone was dissolved in 2 mL acetone. Then edaravone solution was dropped into Eudragit RS 100 solution. This solution was injected into 20 mL water containing 0.1% Pluronic F-68 under moderate magnetic stirring (1000 rpm). Finally, the organic solvents were evaporated by solvent evaporation.

### 4.3. Preparation of Nanocomposite Hydrogel

1.5% (w/w) sodium alginate was added to the nanoparticle suspension under stirring, then dropped 0.5% (w/w) calcium chloride and glycerol to the mixture to obtain edaravone-loaded nanocomposite hydrogel.

### 4.4. Physicochemical Characterization of NP

The hydrodynamic diameter, size distribution, and polydispersity index (PDI) of the NP were measured by dynamic light scattering (DLS) (Omni multi-angle particle size and high sensitivity Zeta potential analyzer, Brookhaven, MS, USA). All DLS measurements were performed at 25 °C with an angle detection of 173° backscatter. For zeta potential measurements by Phase Analysis Light Scattering (PALS) (Omni multi-angle particle size and high sensitivity Zeta potential analyzer, Brookhaven, MS, USA), samples were diluted with ultrapure deionized water (18.2 MΩ.cm). All measurements were performed in triplicate.

### 4.5. Scanning Electron Microscope

After the edaravone-loaded nanoparticles, calcium alginate hydrogel and nanocomposite hydrogel were prepared, the sample was dropped on a silicon wafer and air-dried naturally. Before all samples were observed under a JSM7500F field emission scanning electron microscope (JEOL, Tokyo, Japan), the samples needed to be sprayed gold. 

### 4.6. Determination of Edaravone Encapsulation Efficiency and Drug Loading

Encapsulation efficiency indicates the quantity of the drug entrapped within NP compared with the total amount of initial drug, which can be used to evaluate whether the preparation method is suitable or the polymer material is selected appropriately. In our study, the encapsulation efficiency was measured directly through the method of ultrafiltration centrifugation. Briefly, the drug-loading NP and free drugs were separated by an ultrafiltration centrifugation tube, then the free edaravone and edaravone entrapped in NP detected through high-performance liquid chromatography (HPLC) (Algilent Technologies, Palo Alto, CA, USA) could indirectly obtain the encapsulation efficiency and drug loading of nanoparticles. The encapsulation efficiency (EE) and drug loading (DL) can be described by the following formula: EE = 1 − mf/mt × 100, DL = mt − mf/(mt − mf + mp) × 100%, where mf refers to the free edaravone that is not entrapped in nanoparticles, mt refers to the mass of total drug, and mp refers to the mass of polymer material.

The standard curve of edaravone was established by HPLC assay before the detection. Different concentrations (1 μg/mL, 2 μg/mL, 4 μg/mL, 8 μg/mL, 16 μg/mL, 32 μg/mL, 64 μg/mL, 128 μg/mL, and 256 μg/mL) of edaravone solution are prepared. The chromatographic conditions were column: Chromatographic column-Bioband HP120-C18, 5μm, 150mm × 4.6mm; Acetonitrile: 0.033% trifluoroacetic acid (40:60); temperature: 25 °C; Flow rate: 1.0 mL/min; detection wavelength: 292 nm; sample size: 20 μL. 

### 4.7. In Vitro Release of Edaravone 

As slightly acidic pH is often observed in the wound environment, the release kinetics studies were performed in phosphate buffered saline (PBS, pH 5). Briefly free EDA, EDA-NP, and EDA-NP-gel were suspended in PBS in dialysis bags (average width 34 mm, cut-off 14,000 Da). These dialysis bags were placed in 200 mL PBS (pH 5) at 37 °C and protected from light. The released edaravone was monitored at different time intervals (0.5 h, 1 h, 1.5 h, 2 h, 3 h, 4 h, 6 h, and 9 h) and detected using the HPLC assay as previously described.

### 4.8. Solubility of Edaravone-Loaded Nanoparticles

Edaravone at 0.7 mg was weighed in a tube and 2 mL H_2_O was added with different pHs (pH 3, pH 4, pH 5, and pH 7). The samples were shaken at 25 °C overnight and filtered (pore sizes 0.22 μm) to remove insoluble edaravone, the filtrate was detected by the HPLC. All measurements were performed in triplicate.

### 4.9. In Vivo Wound Healing Assay

All animal experiments were approved by the Third Military Medical Association for Animal Ethics. C57BL/6 male mice aged 8 to 9 weeks were purchased from Sibefu Biotechnology Co., Ltd. (Beijing, China). At 9 to 10 weeks of age, the mice were rendered diabetic by intraperitoneal injection of 50 mg/kg of streptozotocin in 0.05 M sodium citrate (pH 4.5) per day for 5 consecutive days. Mice whose fasting blood glucose level reached ≥ 11.1 mmol/L or random blood glucose levels reached ≥ 16.7 mmol/L twice were used for further study.

All diabetic male C57BL/6J mice were weighed and randomly grouped into five groups (control, EDA-PBS, EDA-NP, EDA-NP-gel (high dose, 0.3 mg edaravone), EDA-NP-gel (low dose, 0.1 mg edaravone)) comprising four animals each, and there was one group of normal male C57BL/6J mice as the other normal control group. The mice were anesthetized with intraperitoneal injections of sodium pentobarbital and the hair area removed was approximately 3 × 3 cm on the back of the mice using a depilatory cream before the surgery. After depilation, the hair was dehydrated with ultrapure water and 75% alcohol to keep skin cleaning and disinfection. On the center of the spine on the exposed skin of the back of the mouse, a round hole with a diameter of 5 mm was made on each side of the neck using a skin biopsy punch, and the full round layer was removed used ophthalmic scissors. Immediately after the wound creation, edaravone solution, EDA-NP, and EDA-NP-gel were applied daily to the wound from day 0 to 6. Mice that were successfully operated on were placed in a single cage at room temperature of 25 °C and given enough food and water. The photographs were recorded on day 0 of the operation, from day 0 to day 13 daily. 

### 4.10. Statistical Analysis

The calculation of the wound area was performed using ImageJ. Statistical analysis was carried out using Prism 7 software (GraphPad Software, Inc., La Jolla, CA, USA). A two-tailed student’s t-test and the Mann–Whitney U-test were used for statistical analysis. *p* < 0.05 was considered statistically significant.

## 5. Conclusions

The main reason for preferring the topical formulations of edaravone is to offer solubility, better bioavailability, and sustained release of edaravone in an active form, which is certainly of great benefit for providing a constant dose of the drug for prolonged periods to improve wound healing. Understanding the perfect dose of edaravone is essential for multiple targets, and above all its complex role in the inflammatory response in wound repair is needed to be addressed before further clinical studies. The present work investigated the combined activity of edaravone and alginate on topical wound healing in diabetic mice. We demonstrated that a low dose of edaravone nanocomposite hydrogel is conducive to wound repair by downregulating ROS levels in diabetic mice close to normal mice. On the contrary, a high dose of edaravone is destructive to wound healing because of insufficient ROS. Thus, the dose might be a key factor in the translational application of edaravone in wound healing. Modulators of free radicals showed promising therapeutic efficacy in in vitro and in vivo studies but found limited success in clinical trials attributable to achieving a subtherapeutic level at the target site [23,24]. The alginate-based nanocomposite hydrogel is promising for diabetic wound healing because of its innate alginate activity and sustained edaravone release.

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
