# Peer review of "Edaravone-Loaded Alginate-Based Nanocomposite Hydrogel Accelerated Chronic Wound Healing in Diabetic Mice"

_marinedrugs, 2019, doi:10.3390/md17050285_

Round 1

Reviewer 1 Report

The manuscript by Fan et al. describes the kinetic improvement in wound healing of diabetic mice.

The results are not clearly described and the captions miss the legend description, which are defined only in the main text.

Three major points weaken the paper:

- in vivo results report some of the same data in three different figures. Data from the same experiments must be presented only once.

- the meaning of negative effect of EDA-NP-gel with high doses of edaravone is only supposed in the discussion. In vitro toxicity tests on the different doses’ formulation should be done to understand the dose dependence or other effects due, for example, to a fast release. Release behavior of the different formulations is also missing.

- efficiency comparison with other types of nanoparticles is only mentioned in the discussion, mentioning edaravone loaded liposomes in ref 12. Comparative experiments should be done to test EDA-NP-gel efficiency vs. other formulations.

Minor points:

- text contain several typos

- English must be improved

- caption improved

- Mentioned Supplementary Figure are not visible to the reviewer

Author Response

Dear Editors and Reviewers:

Thank you for your letter and for the reviewers’ comments concerning our manuscript entitled “Edaravone-loaded alginate-based nanocomposite hydrogel accelerated chronic wound healing in diabetic mice” (ID: marine drugs-492950). Those comments are all valuable and very helpful for revising and improving our paper, as well as the important guiding significance to our researches. We have studied comments carefully and have made corrections which we hope to meet with approval. Revised portion are marked in blue in the paper. The main corrections in the paper and the responds to the reviewers’ comments are as following:

Responds to the reviewers’ comments:

Reviewer #1:

1.     In vivo results report some of the same data in three different figures. Data from the same experiments must be presented only once.

Response: we are very sorry for our incorrect manipulation of those data. We made independent experiments to test wound healing properties in different edaravone-loaded formulations. We had set control group in each experiment, when we analysed the data, we found there is some differences for the same control group in each independent experiment. In order to avoid misunderstanding, in our original version, we had analysed all the control group at the same time to get one data to present it in all independent experiments. We agree to the reviewer, data from same experiments must be presented only once. So in the revised manuscript, we managed the data respect to independent experiment. You may find “diabetic control” and “normal control” in figure 4,5,6 and figure s2 are not exactly same. That is because they are from independent experiments.

2.     The meaning of negative effect of EDA-NP-gel with high dose of edaravone is only supposed in the discussion. In vivo toxicity tests on the different doses’ formulation should be done to understand the dose dependence or other effects due, for example, to a fast release. Release behavior of the different formulatios is also missing.

Response: in fact we had performed wound healing effects of high and low dose of edaravone as presented in figure ?and figure? Concerning the in vitro toxicity, edaravone has been approved in Japan as a drug to treat acute-phase cerebral infarction, and in 2015 it was approved for amyotrophic lateral sclerosis (ALS). In 2017, the FDA also approved edaravone for treatment of patients with ALS [1]. In all approved case, edaravone was applied by intravenous injection, which means the safety of edaravone has been validated [2]. In our work, we propose edaravone for topical application, which is less challenging concerning safety.

We are grateful for reviewer’s suggestion to test the release behavior of high and low dose of edaravone nanocomposite hydrogel. We have set additional experiment to study the release profile of high and low dose of edaravone loaded nanocomposite hydrogel. Even their release profiles are quit the same concerning percentage of edaravone released, as shown as following figure 1(figure s3). However high dose of edaravone loaded nanocomposite hydrogel released almost 3-fold of edaravone than low dose of edaravone (figure 1). Probably the large amount of edaravone released by EDA-NP-gel depleted too much ROS in the wounds, which broken the homeostasis, to hamper the healing process.

3.     Efficiency comparison with other types of nanoparticles is only mentioned in the discussion, mentioning edaravone loaded liposomes in ref 12. Comparative experiments should be done to test EDA-NP-gel efficiency vs other formulations.

Response: In fact we compare encapsulation efficiency of our system with other edaravone formulation, because we would like to prove our system is suitable for edaravone loading and delivery. The encapsulation efficiency is comparable parameter, like size and zeta potential, between different formulations. The edaravone liposomes improved the bioavailability of edaravone due to the loading efficiency and protection effect of liposomes. Similarly, our nanocomposite hydrogel can load edaravone and protect it from unwanted factors, so we proved our system is adaptable for edaravone application.

 In addition, the edaravone liposomes were used to protect against light-induced retinal damage in mice not promote wound healing. In our opinion, we can not compare pharmacological activities of those two delivery systems since they were developed for different applications. To our best knowledge, there are not any edaravone related nanosystems used to treat chronic wound. Thus we only discuss the comparable parameters of our system with other formulations.

4.     Minor points: text contain several typos; English must be improved; caption improved; mentioned supplementary figure are not visible to the reviewer.

Response: We are very sorry about the minor mistakes mentioned by reviewer. We have corrected the typos, caption and improved the English as much as we can. We also prepared a pdf version of supplemental data to make sure reviewers can check it without difficulties.

Special thanks for your good comments!

Reviewer 2 Report

In this paper, authors presented an alginate-based hydrogel, loaded with edavarone encapsulated in eudragit nanoparticles, for the treatment of chronic wound healing.

This article is well structured and conclusions are supported by the presented results. I suggest the acceptance of this paper after minor revision. I also suggest a spell check of English.

General comments:

Page 2, line 48: delete “in”;

Page 2, line 67: please modify with “novel edaravone-loaded..”

Page 2, line 73: please correct “therapeutics”

Figure 1: please add in the caption the complete name off “ERL”;

Table 1: I suggest to add to this table also the data of Drug Loading.

Figure 2: it is not clear the statistical difference between groups. The statistic symbols could be placed next to the legend; or the statistical analysis results could be described with a sentence in the main text.

Author Response

Dear Editors and Reviewers:

Thank you for your letter and for the reviewers’ comments concerning our manuscript entitled “Edaravone-loaded alginate-based nanocomposite hydrogel accelerated chronic wound healing in diabetic mice” (ID: marine drugs-492950). Those comments are all valuable and very helpful for revising and improving our paper, as well as the important guiding significance to our researches. We have studied comments carefully and have made corrections which we hope to meet with approval. Revised portion are marked in blue in the paper. The main corrections in the paper and the responds to the reviewers’ comments are as following:

Responds to the reviewers’ comments:

Reviewer #2:

1.     Page 2, line 48: delete “in”

2.     Page 2, line 67: please modify with “novel edaravone-loaded..)

3.     Page 2, line 73: please correct “therapeutics”

4.     Figure 1: please add in the caption the complete name of ERL

Response: we are very sorry for our incorrect English spelling. We have made our corrections.

5.     Table 1 I suggest to add to this table also the data of drug loading

Response: considering the reviewer’s suggestion, we have add “drug loading” in table 1, which is 9.25 ± 0.25%(w/w).

6.     Figure 2: it is not clear the statistical difference between groups. The statistic symbols could be placed next to the legend or the statistical analysis results could be described with a sentence in the main text.

Response: It is really true as reviewer suggested that we should present the statistical analysis clearly, so we described the results with a sentence in the main text of caption.

Very special thanks for your good comments!

We tried our best to improve the manuscript and made some changes in the manuscript. These changes will not influence the content and framework of the paper. Here we did not list the changes but marked in blue in revised paper.

We appreciate for editors and reviewers’ warm work earnestly, and hope that the correction will meet with approval.

Once again, thank you very much for your comments and suggestions.

Round 2

Reviewer 1 Report

The authors improved the paper and adequately addressed the reviewer’s questions.